# Study on Mechanical Properties and Erosion Resistance of Self-Compacting Concrete with Different Replacement Rates of Recycled Coarse Aggregates under Dry and Wet Cycles

**Shan Liu [1], Fengxia Han [1,2,*], Shiqi Zheng [1], Songpu Gao [1] and Guoxing Zhang [1]**

1   College of Architectural and Civil Engineering, Xinjiang University, Urumqi 830047, China; liushan@xju.edu.cn (S.L.); 107552101534@stu.xju.edu.cn (S.Z.); gaosongpu@stu.xju.edu.cn (S.G.); zhangguoxing1@stu.xju.edu.cn (G.Z.)
2   Key Laboratory of Building Structure and Seismic Resistance of Xinjiang, Urumqi 830017, China
*   Correspondence: fxhan@xju.edu.cn

**Abstract:** Concrete that self-compacts is frequently utilized in engineering construction. Recycled coarse aggregate self-compacting concrete (RCASCC) is made by partially substituting recycled coarse aggregates (RCA) for natural coarse aggregates in order to conserve construction resources. This study examines the impact of linked sulfate erosion, dry and wet cycles, and RCA replacement rates of 0%, 25%, 50%, 75%, and 100% on the mechanical properties and durability of RCASCC. By using the mass loss rate, relative dynamic elastic modulus, corrosion resistance factor, X-ray diffraction (XRD), scanning electron microscope (SEM), and atomic force microscope (AFM) analyses, as well as other macroscopic and microscopic methods, it is possible to examine the deterioration patterns of RCASCC under dry and wet cycles. The results demonstrate that the addition of RCA has a notable impact on concrete's resistance to sulfate attack during both dry and wet cycles. The erosion products steadily rise, the interfacial transition zone (ITZ) becomes rougher, and the sulfate resistance falls as the replacement rate of RCA rises. According to the findings of $SiO_2$, AFt, and $CaCO_3$, the examination of corrosion products from XRD and microstructure from SEM and EDS is carried out. The old mortar that has adhered to the surface of RCA, as shown by the AFM analysis of ITZ and the SEM analysis of RCA, can significantly affect the roughness of ITZ inside RCASCC.

**Keywords:** recycled coarse aggregate; self-compacting concrete; dry and wet cycles; sulfate resistant; interface transition zone (ITZ)



## 1. Introduction

An enormous hazard to the human ecological environment is posed by the destruction of obsolete buildings in China, which generates a lot of construction solid waste, of which concrete solid waste makes up around 40% [1]. Instead of natural coarse aggregate (NCA), many researchers have employed recycled coarse aggregate (RCA). The "Standards for the Use of Recycled Aggregates and Recycled Aggregate Concrete" were recommended by the Japan Construction Industry Association in 1977, and recycled aggregates and recycled aggregate concrete were defined in Chapter 4. When the replacement rate of the recycled coarse aggregate was 50%, the concrete's strength was enhanced by readjusting the aggregate's particle gradation. Additionally, there was an increase in diversity in concrete strength.

In 1992, Ozawa proposed a self-compacting concrete substance. Flow capacity (filling), passage capacity (fluidity), viscosity, and stability (resistance to segregation) are the characteristics of self-compacting concrete [2–7]. The mechanical properties of recycled aggregate self-compacting concrete do not significantly alter when the percentage of recycled aggregate does not exceed 50% [8–11]. Numerous studies have demonstrated that with the right replacement ratio, recycled coarse aggregate self-compacting concrete (RCASCC)

can perform better than standard self-compacting concrete in terms of workability and fundamental mechanical properties [12–16]. Thus, it is possible to use RCASCC.

Sulfate ions have the potential to seriously harm concrete structures. Gypsum and calcium sulfo-aluminate are produced when sulfates react with the calcium silicate hydrated (C-S-H) gel, expanding in volume and causing surface cracking or softening. Additionally, cracks facilitate the infiltration of corrosive water containing sulfate and other ions, which speeds up the decomposition of the concrete. Cracks also damage the cement hydrates' bonding characteristics, which significantly reduces the durability of concrete [17]. Recent studies have looked at recycled concrete with coarse particles' resilience to sulfate. While the results vary, they all demonstrate that high replacement rates lower a material's resistance to sulfate attack. These investigations used recycled aggregates with various compositions as well as various substitution rates and erosion settings [17]. Despite substantial research on the mechanical and durability characteristics of SCC produced using RCA, only a small number of studies have specifically focused on these characteristics [18].

Many locations in Xinjiang are salty, and salty soil includes a variety of hazardous ions, the most prevalent of which are sulfate ions [19]. Few investigations on the dry and wet sulfate cycle erosion of recycled aggregate self-compacting concrete have been carried out thus far. The current study expands on previous research by examining the impact of recycled aggregate self-compacting concrete's sulfate resistance in order to provide some reference indices for its practical implementation [9]. A reference index for the creation of requirements for self-compacting concrete with recycled aggregate is another goal of the project. Additionally, it serves as a reference for the creation of self-compacting concrete requirements [20]. By integrating experimental approaches and theoretical analysis, the damage process and degradation mechanism of self-compacting concrete with RCA during a dry–wet cycle and linked sulfate erosion are explored [14].

## 2. Materials and Methods

### 2.1. Materials

The cement used was Xinjiang-produced Tianshan P.O42. 5 ordinary silicate cement. This test made use of Class II fly ash created by Urumqi West Construction. The continuous grading natural pebble with a particle size of 5–20 mm was known as natural coarse aggregate (NCA). Small- and medium-sized waste concrete from construction solid waste is broken by a jaw crusher and sieved through automatic screening equipment to generate a target particle size of 5–20 mm. RCA was made from local building solid waste. Table 1 displays the fundamental mechanical characteristics of RCA and NCA. The water was Urumqi's household water supply. In this test, 10% sulfate solution was employed to simulate concrete erosion since northwest China has more saline soils and concrete is more badly eroded by sulfate. The type of additive chosen was a polycarboxylic-acid-type water-reduction agent, whose water-reducing efficiency was 21%. Northwest China has more saline soils and the concrete is more seriously eroded by sulfate, 10% sulfate solution was used to simulate the erosion in this test [21].

**Table 1.** Basic physical properties of recycled coarse aggregates and natural coarse aggregates.

| Type of Coarse Aggregates | Water Absorption (%) | Apparent Density (kg/m³) | Needle and Flake Content (%) | Solidity (%) | Micronized Content (%) | Crush Index (%) |
|---|---|---|---|---|---|---|
| Recycled Coarse Aggregates | 4.89 | 2450 | 6 | 7.5 | 1.4 | 12.8 |
| Natural Coarse Aggregates | 0.54 | 2700 | 3 | 1.0 | 1.0 | 3.87 |

### 2.2. Concrete Mix Ratio

In this test, C30 self-compacting concrete was created according to the standard with 0%, 25%, 50%, 75%, and 100% of RCA replacement. The RCASCC formula was discovered

through the subject group's study, and it served as the foundation for the ratio design, as illustrated in Table 2.

**Table 2.** Concrete mix.

| Type | Water (Kg) | Additional Water (Kg) | Rock (Kg) | Recycled Aggregates (Kg) | Sand (Kg) | Cement (Kg) | Fly Ash (Kg) | Additives (Kg) |
|------|-----------|----------------------|-----------|--------------------------|-----------|-------------|--------------|----------------|
| R-0 | 6.13 | 0 | 30.52 | 0 | 26.17 | 9.27 | 9.35 | 0.12 |
| R-25 | 6.13 | 0.5 | 22.8 | 7.3 | 26.17 | 9.27 | 9.35 | 0.12 |
| R-50 | 6.13 | 1 | 15.25 | 14.5 | 26.17 | 9.27 | 9.35 | 0.12 |
| R-75 | 6.13 | 1.48 | 7.61 | 21.5 | 26.17 | 9.27 | 9.35 | 0.12 |
| R-100 | 6.13 | 1.98 | 0 | 28.9 | 26.17 | 9.27 | 9.35 | 0.12 |

Note: R-0, R-25, R-50, R-75, and R-100 indicate the 0%, 25%, 50%, 75%, and 100% recycled aggregate replacement rate of self-compacting concrete test blocks, respectively.

### 2.3. Working Performance

Slump expansion and expansion time were used to characterize the fillability of self-compacting concrete in accordance with the determination methods and specifications. After the slump expansion is over, the result of the J-ring expansion measurement should be the average of two perpendicular diameters of the expansion surface. The segregation rate sieve test was used to gauge resistance to segregation [22]. The results are shown in Table 3.

**Table 3.** Working performance.

| Type | Slump Expansion (mm) | Extension Time (s) | J-Ring Extension Expansion (mm) | Resistance to Segregation (%) |
|------|---------------------|--------------------|--------------------------------|-------------------------------|
| RCASCC-0 | 620 | 2.6 | 4 | 8.9 |
| RCASCC-25 | 650 | 2.3 | 2 | 9.9 |
| RCASCC-50 | 610 | 2.9 | 8 | 7.5 |
| RCASCC-75 | 590 | 3 | 10 | 4.1 |
| RCASCC-100 | 600 | 3.1 | 9 | 3 |

Note: RCASCC-0, RCASCC-25, RCASCC-50, RCASCC-75, and RCASCC-100 indicate the 0%, 25%, 50%, 75%, and 100% recycled aggregate replacement rate of self-compacting concrete test blocks, respectively.

### 2.4. Mechanical Properties

The 28d cube compressive strength of the RCASCC was calculated for five replacement rates, and the fundamental mechanical properties of the concrete were tested in accordance with the Standard for test procedures of concrete physical and mechanical properties. The compressive strength of the RCASCC was tested by the WHY-3000 automatic pressure testing equipment using 100 mm × 100 mm × 100 mm cube specimens with three specimens per group as Figure 1. The results were averaged.

### 2.5. Durability

The NELD-LSC-type concrete sulfate dry and wet cycle machine is used for this test, and each dry and wet cycle lasts a total of 22 h in accordance with the standard. Each wet and dry cycle takes 22 h in total. When the mass loss rate hits 5%, the corrosion resistance factor of compressive strength reaches 75%, or 120 wet and dry cycles of the intended sulfate resistance level have been completed, the test may be terminated. Three specimens from each group of the 100 mm × 100 mm × 400 mm prism and 100 mm × 100 mm × 100 mm cube were tested for durability throughout a 28-day curing period [23].

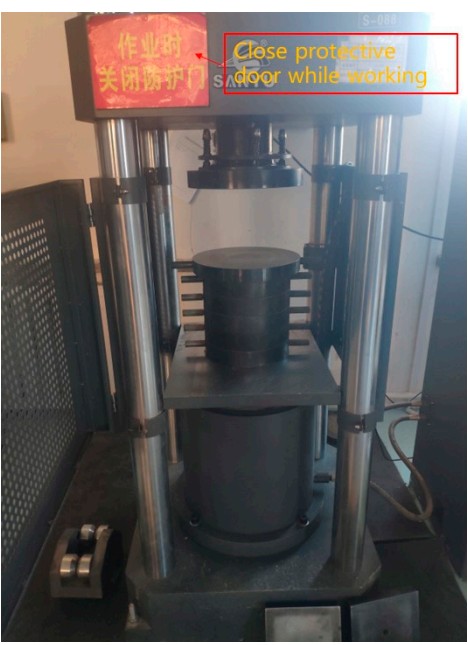

**Figure 1.** WHY-3000 automatic pressure testing machine.

*2.6. Corrosion Resistance Factor*

The ratio of the measured compressive strength of a group of concrete specimens subjected to sulfate attack after dry and wet cycles N times to the measured compressive strength of a group of concrete specimens subjected to sulfate attack specimens of the same age of standard curing is known as the concrete compressive strength corrosion resistance factor [24–26]. Equation (1) illustrates the precise formula. The value can be used to react to the various ages of erosion of concrete specimens during the wet and dry cycles of sulfate erosion value changes [9].

$$K_f = \frac{f_{cn}}{f_{c0}} \times 100\%,\qquad(1)$$

$K_f$—the coefficient of the corrosion resistance of compressive strength (%).
$f_{cn}$—the measured compressive strength of a group of concrete specimens subjected to sulfate attack after N wet and dry cycles (MPa), accurate to 0.1 MPa.
$f_{c0}$—the measured value of the compressive strength (MPa) of a group of standard cured concrete specimens subjected to sulfate attack at the same age as the specimens, accurate to 0.1 MPa.

*2.7. Microscopic Test*

The samples of the microscopic test were chosen for SEM, XRD and AFM examination after 120 dry and wet cycles of sulfate erosion. SEM was tested by TESCAN MIRA LMS, XRD was tested by Rigaku SmartLab SE, AFM was tested by Bruker Dimension Icon.

**3. Analysis of Experimental Results**

*3.1. Basic Mechanical Properties*

The compressive strength results are displayed in Table 4 and Figure 2. The compressive strength at 7 days and 28 days revealed a tendency of initially increasing and then declining, and the replacement rate of RCA ranged from 0% to 100%. The greatest compressive strengths of RCASCC at 7 and 28 days were 27.9 MPa and 35.9 MPa, respectively, when the recycled aggregate substitution rate was 50%. The 7/28-day strength ratio grew and subsequently declined as the replacement ratio for RCA increased, showing that the addition of RCA boosted the early strength but did not improve the latter strength. At 50% of the RCA replacement rate, the greatest contribution to early strength improvement

was found. The primary cause is the presence of recycled coarse aggregate (RCA) in the cement paste, which has a negative impact due to the index of its crusading strength, and a positive effect due to the pre-wetting treatment of RCA in the curing process of the surrounding cement, which will play a role in the internal curing and at the same time increase the density of the surrounding part of the concrete. The recycled coarse aggregate's large crushing index and low hardness are caused by some cement paste, which has the unfavorable consequence of lowering the strength of the concrete as a whole [27]. The aggregate gradation is good when the replacement ratio is 50% since the benefits of recycled coarse aggregate outweigh the drawbacks.

**Table 4.** Compressive strength.

| Type | 7d$f_{cu}$ (Mpa) | 28d$f_{cu}$ (Mpa) | Elastic Modulus/Mpa |
|---|---|---|---|
| RCASCC-0 | 24.1 | 32.1 | 29.8 |
| RCASCC-25 | 24.2 | 31.9 | 29.3 |
| RCASCC-50 | 27.9 | 35.9 | 28.9 |
| RCASCC-75 | 19.2 | 27.8 | 27.6 |
| RCASCC-100 | 17.5 | 24.6 | 26.6 |

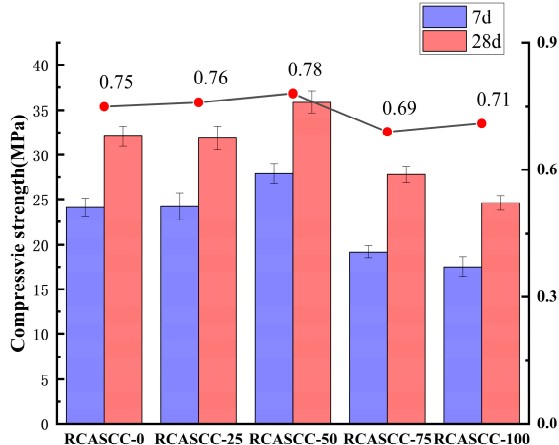

**Figure 2.** Compressive strength with different RCA replacement ratios and 7/28-day strength ratio.

The most notable improvement in early and late strength was therefore shown when 50% RCA was added in place of NCA [28].

### 3.2. Exterior Erosion Characteristics

All concrete prismoids in this section were examined to determine the degree of damage after the exposure to the sulfate solution was prolonged for up to four months (120 cycles of wetting and drying).

Figure 3 depicts how the RCASCC changed following erosion. Figure 3a,b demonstrate that the amount of white material on the specimen's surface increased, and mortar and aggregate spalled on the prism's side and corners [9]. The graphs in Figure 3a–e demonstrate that $SO_4^{2-}$ erosion increased with the RCA substitution rate. The most significant damage was found in RCASS, where RCA entirely replaced NCA. The mortar layer's topmost portion had entirely peeled off, there were visible cracks, and the aggregate exposure issue was critical. Figure 2 shows that the self-compacting concrete with more than 50% RCA substitution was prone to $SO_4^{2-}$ erosion to the greatest extent. This finding was primarily supported by the most obvious cracking and spalling of the specimens as a whole, as well as the most erosion products on the specimens' surfaces [29].

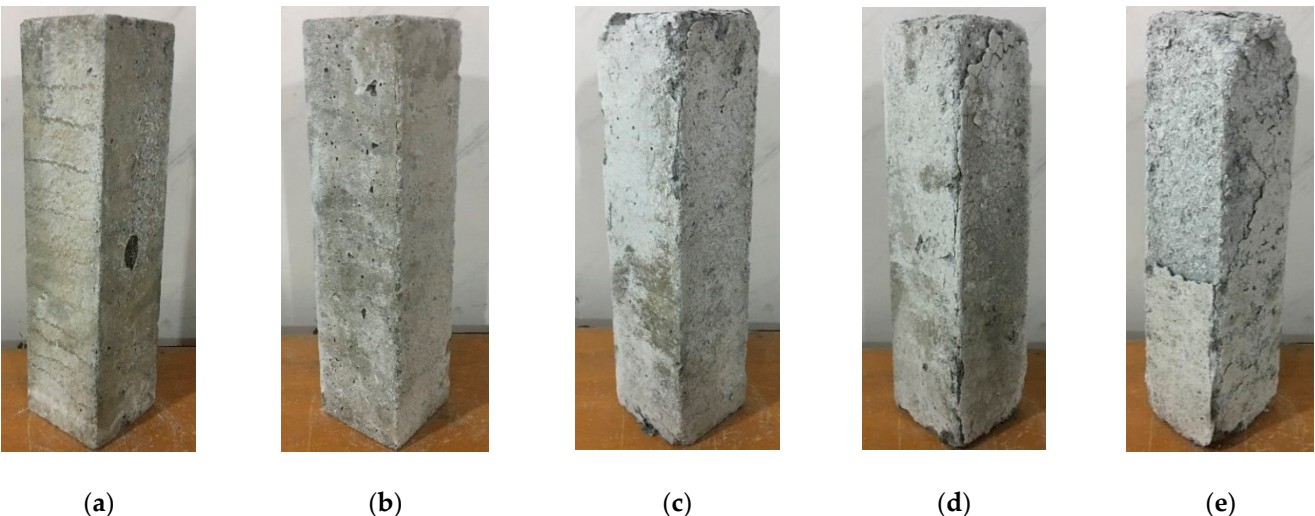

(**a**)　　　　　　(**b**)　　　　　　(**c**)　　　　　　(**d**)　　　　　　(**e**)

**Figure 3.** Appearance of prismatic specimens after erosion. (**a**) RCASCC-0. (**b**) RCASCC-25. (**c**) RCASCC-50. (**d**) RCASCC-75. (**e**) RCASCC-100.

*3.3. Mass Loss Rate*

Figure 4 displays the mass loss rate of self-compacting concrete with various RCA replacement rates. The pattern persisted along all curves. RCASCC demonstrated a change in the law of initially increasing and then dropping as the number of wet and dry cycles increased. The mass loss rate of each set of test specimens varied after the initial 30 wet and dry cycles. Due to the reaction of $SO_4^{2-}$ in the first solution with the hydrates in the concrete, which produced compounds such as calcium alumina, which caused it to plug some of the pores in the concrete and improve the compactness, RCASCC-50 showed the biggest increase in mass damage rate of 1.68%. Because the larger recycled aggregate replacement rate reflects a larger porosity of the concrete, allowing the products from sulfate attack to continue to be stored in the concrete, the mass loss rate continued to increase for RCASCC-75 and RCASCC-100 when the number of cycles reached 60 while decreasing for the other groups. The specimens' overall mass loss rate began to reduce after more than 60 wet and dry cycles. After a sudden increase, the rate of mass loss for the specimens in the RCASCC-50, RCASCC-75, and RCASCC-100 groups reduced to 2.98%, 3.05%, and 3.59%, respectively.

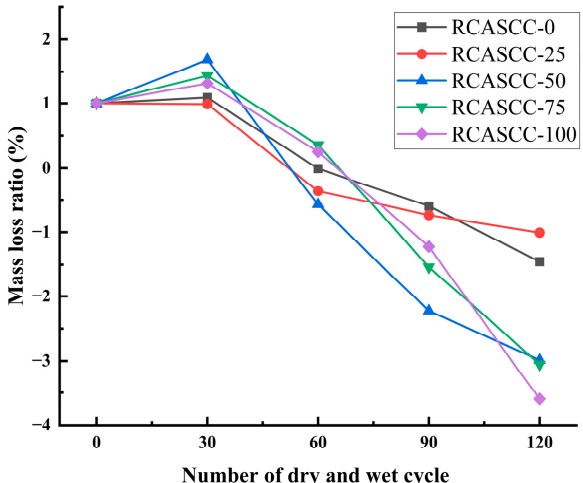

**Figure 4.** Mass loss rate of self-compacting concrete with recycled aggregates.

Different RCA replacement rates can significantly affect the sulfate attack resistance of self-compacting concrete, as shown in Figure 4. The specimen's total water content and porosity increased when excessive RCA was applied in place of NCA. Increased amounts of $SO_4^{2-}$ were absorbed into the concrete. As a result, sulfate attack significantly increased, which was bad for the concrete's ability to resist sulfates [30,31].

### 3.4. Relative Dynamic Elastic Modulus

The relative dynamic modulus of elasticity of self-compacting concrete with RCA varies at various replacement rates, as shown in Figure 5. Similar variations in the relative dynamic elastic modulus of self-compacting concrete were observed with various RCA replacement rates. The relative dynamic elastic modulus exhibits an overall "increase first, then decrease" fluctuation pattern due to the rise in the number of dry and wet cycles and the erosion of the sulfate coupling. Throughout the wet and dry cycles, the relative dynamic elastic moduli decreased by 76.3%, 80.9%, 73.1%, 61.2%, and 59.6% for each group. RCASCC-75 and RCASCC-100, which neared destruction, experienced larger relative dynamic elastic modulus drops. For this phenomenon, RCA has a higher porosity than NCA, which affects the rate of $SO_4^{2-}$ erosion while also increasing the contact area between $SO_4^{2-}$ and the cement's hydration products, improving the rate at which calcite and other products are produced, filling internal pores, increasing specimen density, and enhancing relative dynamic elastic modulus [9]. Its expansion stress keeps increasing as calcium alumina and other hydration products are produced and stored, which accelerates the development of pores and microcracks [32,33]. As a result, the specimen's internal structure is destroyed, and eventually, the damage spreads to the surface of the specimen, causing the relative dynamic elastic modulus to fail [34]. RCASCC-0 saw the lowest relative dynamic elastic modulus drops, followed by RCASCC-25 and RCASCC-50. RCASCC-0, RCASCC-25, and RCASCC-50 all had relative dynamic elastic moduli that were greater than 70%, and they all maintained a high level of resistance to sulfate erosion. This result demonstrated that the relative dynamic modulus of elasticity of self-compacting concrete was significantly influenced by various RCA substitution rates. The relative dynamic modulus of elasticity of self-compacting concrete changed in a manner that was more impressive the greater the replacement rate of RCA [35].

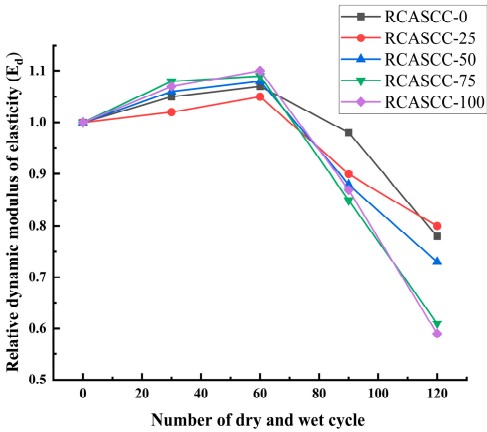

**Figure 5.** Relative dynamic modulus of elasticity of self-compacting concrete with recycled aggregates.

### 3.5. Corrosion Resistance Factor

Corrosion resistance factors of self-compacting concrete with RCA under the various phases of dry and wet cycles of sulfate erosion were discovered based on the cubic compressive strength values, as shown in Table 5. Then, a plot of their variation laws was made. Figure 6 demonstrates how the early stages of the modest number of dry and wet cycles of sulfate erosion did not exhibit any variations in the compressive strength of self-compacting concrete with varying doses of recycled aggregates. After 30 cycles of dry

and wet conditions, the first four groups displayed modest increases with growth rates of 1.1%, 2.5%, 3.9%, and 1.8%. In the later stage, all displayed a declining tendency, but the decline was more consistent. Generally speaking, the deterioration got worse as there were more dry and wet cycles of sulfate erosion. The corrosion resistance factor for the fifth group of specimens consistently displayed a downward trend, indicating that their compressive strength gradually declined.

**Table 5.** Compressive strength with different numbers of dry and wet cycles.

| Number of Cycles | 30 | 60 | 90 | 120 |
| --- | --- | --- | --- | --- |
| RCASCC-0 | 29.43 | 27.54 | 27.19 | 26.93 |
| RCASCC-25 | 29.04 | 27.99 | 26.91 | 25.86 |
| RCASCC-50 | 34.55 | 32.55 | 31.95 | 30.72 |
| RCASCC-75 | 25.14 | 23.56 | 21.40 | 20.88 |
| RCASCC-100 | 22.89 | 21.92 | 21.16 | 20.06 |

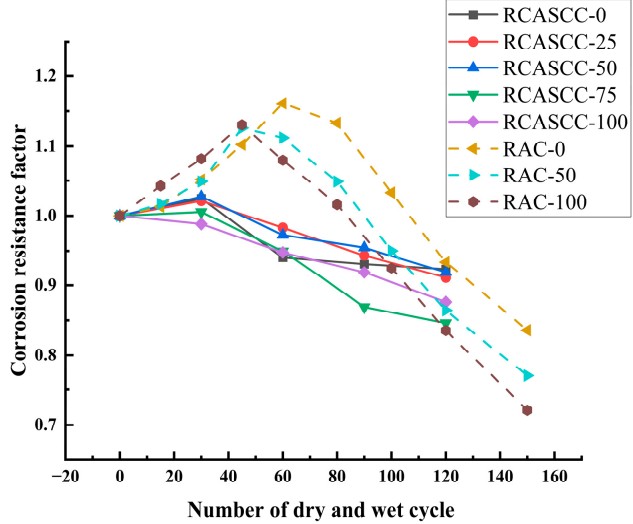

**Figure 6.** Corrosion resistance factor of recycled aggregate self-compacting concrete.

The initial erosion products that were produced when the specimens were first eroded by $SO_4^{2-}$ filled the concrete pores, increasing mass and creating a denser interior structure. Higher compressive strength was readily evident from the exterior results. The quantity of erosion products rose and the pore structure was continually filled as the erosion reaction went on. The interior pores crossed the stress threshold, accompanied by the tendency of certain calcium alumina and other compounds to self-expand. The specimen's overall structure started to deteriorate progressively as cracks started to appear and grow, which reduced compressive strength [36]. The specimens from the RCASCC-50 group had a significantly better corrosion resistance factor than those from the other four groups, which explains their high compressive strength all around. The good benefit outweighed the negative effect when the RCA replacement rate was 50%. The transmission of $SO_4^{2-}$ can also be somewhat hampered when the internal concrete density is high and the aggregate particle gradation is better. As a result, the specimens in the RCASCC-50 group had greater corrosion resistance factors.

Figure 6 depicts the variation pattern of the corrosion resistance factor of recycled aggregate concrete (RAC) in both dry and wet environments of sulfate [37,38]. The results of this paper likewise displayed a pattern of rising and then falling, which fit the pattern. However, the corrosion resistance factor's greatest peak value looked to be different. In this study, sodium sulfate and the highest corrosion resistance factor were used to create a 10% sulfate solution, and around 30 dry and wet cycles were performed. Figure 6 displays

the greatest corrosion resistance factor and a 5% sulfate solution made with magnesium sulfate. There were perhaps 60 dry and wet cycles in total. The fact that the components used to prepare the solution and the sulfate concentration varied could be one explanation; as a result, the concrete was vulnerable to varying degrees of erosion effect [39].

### 3.6. Internal Damage Mechanism

Five sets of samples were chosen for XRD, SEM, and AFM examination based on the macroscopic test findings and their representativeness after 120 dry and wet cycles of sulfate erosion, as shown in Figure 7.

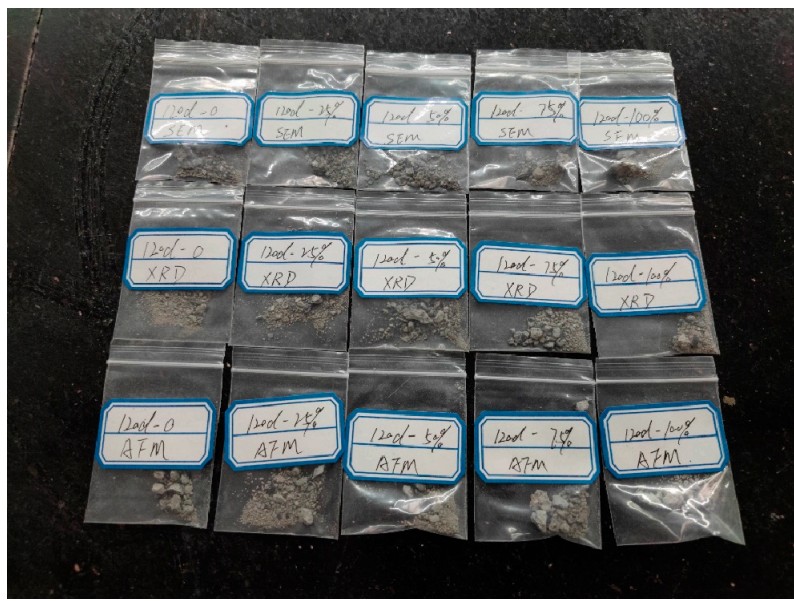

**Figure 7.** Microscopic test samples.

### 3.6.1. Changes in the Content of Each Element in Concrete

The specimens' XRD test results after 120 dry and wet cycles are shown in Figure 8. $SiO_2$, $Ca(OH)_2$, $CaCO_3$, $CaO\bullet Al_2O_3$, and somewhat weak calcium alumina (Aft) diffraction peaks were the most prominent in the RCASCC specimens for the five fitting ratios. In concrete specimens with various fit ratios, the distribution of the overall compounds was more obvious, and the relative concentration could be inferred from the size of the peaks. Both specimens from (a) and (b) contained Aft. The matching diffraction peaks, however, were smaller. The specimens with a lower replacement rate of RCA demonstrated better resistance to sulfate attack from both the internal structure and the external surface, along with lighter external damage and a smaller decrease in the relative dynamic elastic modulus of the specimens in the macroscopic experiments [25,40]. Groups (c), (d), and (e) of the $CaO\bullet Al_2O_3$ diffraction peak were all rather faint. The relative dynamic elastic modulus later results could, nevertheless, correspond to the findings that the later relative dynamic elastic modulus declined more when the substitution rate of RCA was higher because the Aft diffraction peaks were much enhanced [26]. The primary cause was that more old mortar adhered to the surface of the aggregate due to a greater RCA replacement rate would further react with $SO_4^{2-}$ with a specific level of activation, leading to substantially more erosion products. The amount of Aft steadily rose together with the growing erosion of $SO_4^{2-}$, and this substance was the primary factor in the specimens' decreased strength. Concrete's mass loss rate, dynamic elastic modulus, and corrosion resistance factor drop were all more pronounced when its amount was high.

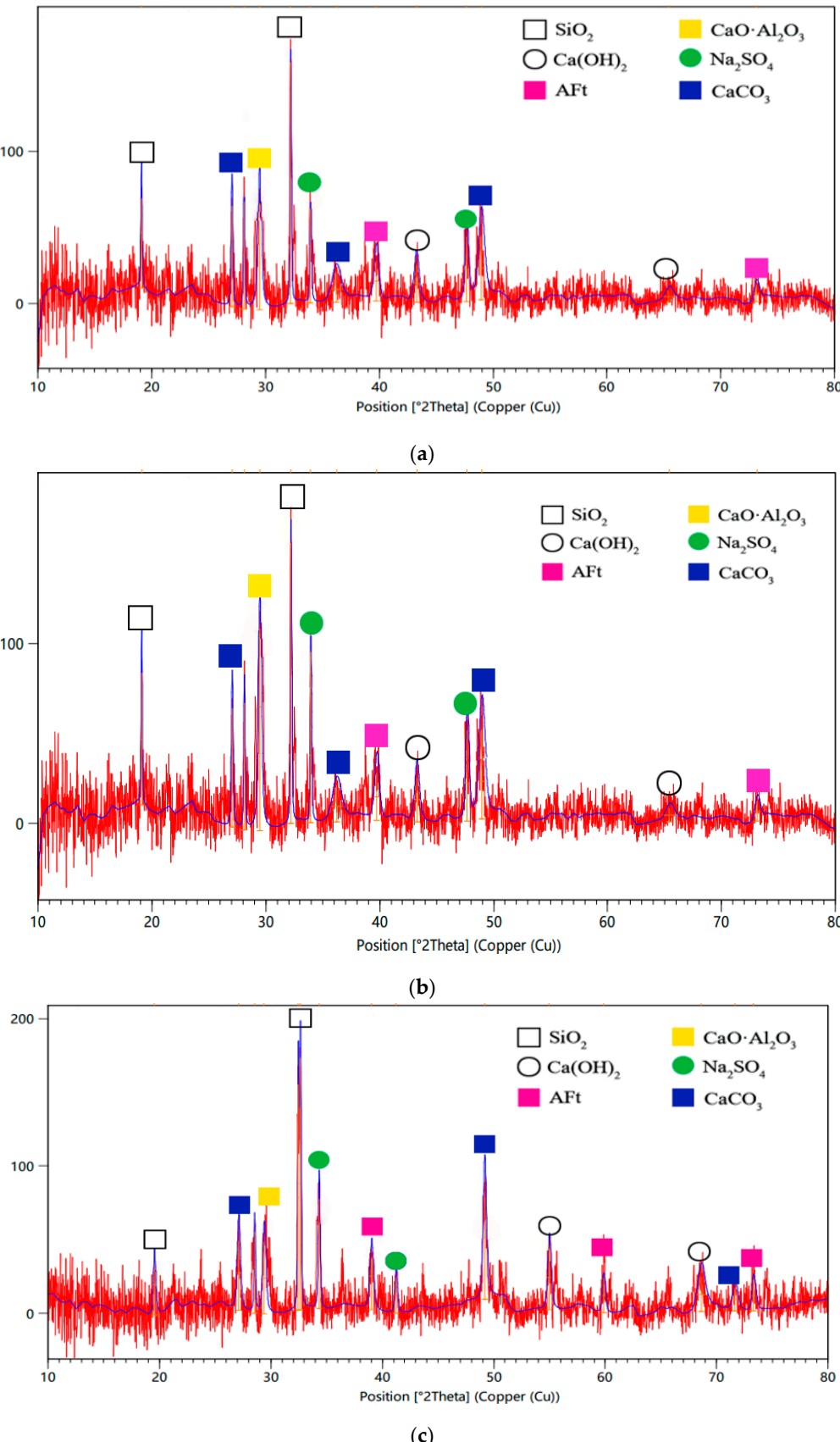

(**a**)

(**b**)

(**c**)

**Figure 8.** *Cont*.

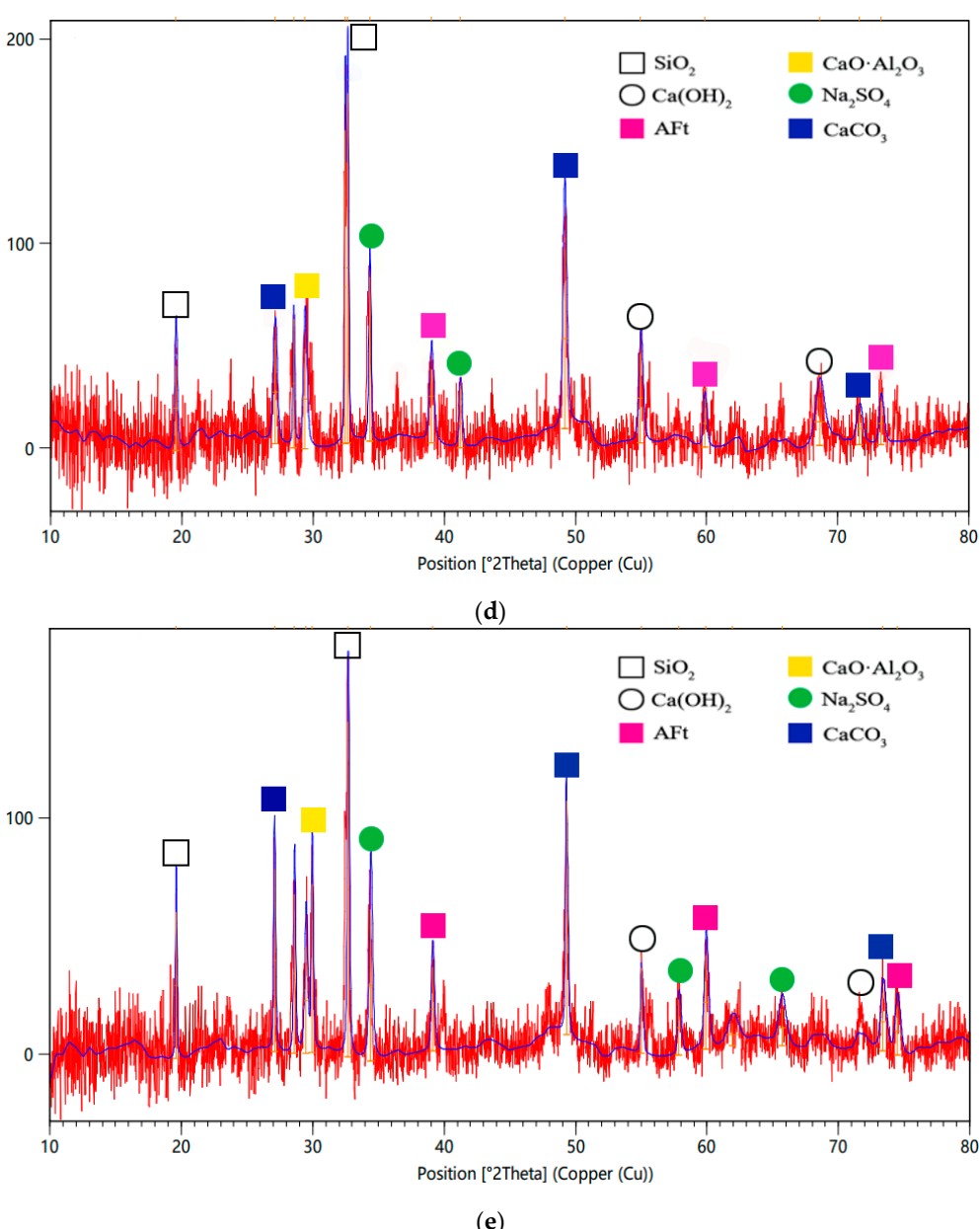

**Figure 8.** XRD results of RCASCC at the end of 120 wet and dry cycles. (**a**) RCASCC-0. (**b**) RCASCC-25. (**c**) RCASCC-50. (**d**) RCASCC-75. (**e**) RCASCC-100.

3.6.2. Mechanisms of Erosion of the Internal Structure of Concrete

Figure 9 demonstrates how columnar and quasi-spherical particles, which were stacked in a specific way to produce a flocculent structure, were the lowest structural units of C-S-H gel at the early stage of hydration. At a later stage, pores and gel pores predominated. These pores appeared as spherical particles and closely bound the sand and aggregate. After 120 dry and wet cycles of sulfate erosion, the RCASCC formed during the hydration reaction of large $Ca(OH)_2$ is shown in Figure 10. The concrete's interior structure was fragile, and cracks were readily visible through the pores. Figure 11 depicts a portion of the pores that were filled with many needle clusters of corrosion products, or Aft. Further examination of various internal features of the specimen revealed several needle clusters and needle columnar Aft accumulation (Figure 12a). The internal structure of the concrete became looser as a result of the squeeze-up effect, which caused internal cracks in the concrete and increased the pore structure. When the internal crowding action of the concrete is combined with the analysis findings of the appearance evolution, the concrete

will crack. As the fissures widened, mortar and aggregates on the exterior surface began to spall. The SEM results of the self-compacting concrete specimens with a high replacement of RCA indicated a considerable amount of Aft, which means that the number of erosion products and the effect of erosion by $SO_4^{2-}$ products were both more pronounced [41]. The XRD diffraction peak intensity analysis, which related to $SiO_2$, Aft, and $CaCO_3$ products in XRD, also showed the relative amounts of erosion products. This observation revealed the precise form and distribution of the compounds and further confirmed their existence in the SEM. Figure 13a,b depict the microscopic morphology of the recycled coarse aggregate self-compacting concrete surface, where the surface roughness varied and the erosion of the outer surface was more severe because of the continuous diffusion of $SO_4^{2-}$. The surface revealed a small amount of spherical sodium sulfate crystals and needle-fine calcium alumina was semi-exposed on the outer side, which more visibly showed the gradual erosion from the outside to the inner surface.

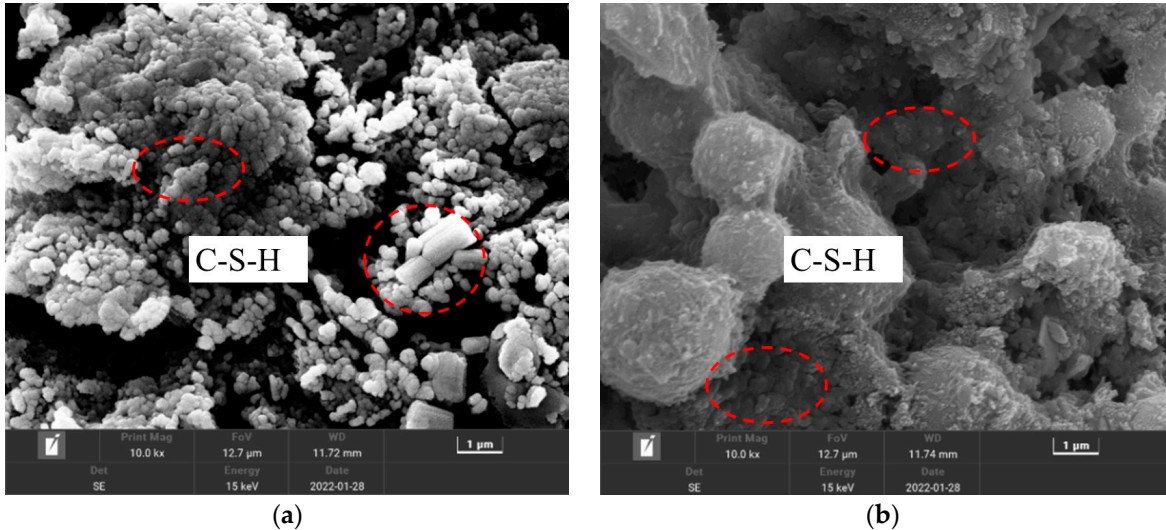

**Figure 9.** SEM results of RCASCC-0 at the end of 120 wet and dry cycles. (**a**) Early stage of C-S-H hydration. (**b**) C-S-H late stage of hydration.

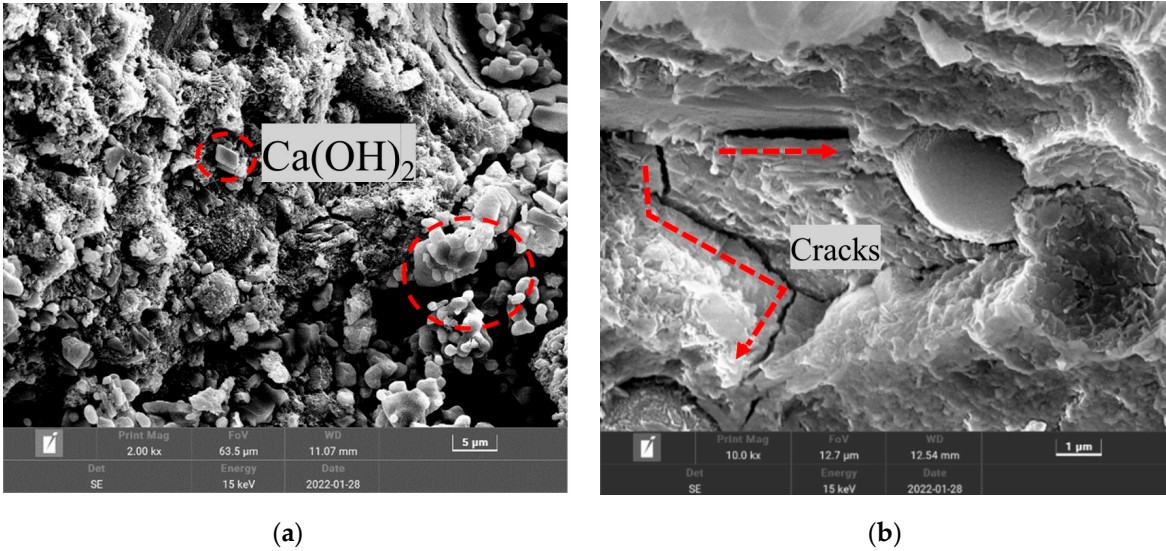

**Figure 10.** SEM results of RCASCC-25 after the end of 120 wet and dry cycles. (**a**) Hydration product $Ca(OH)_2$. (**b**) Internal structural cracks.

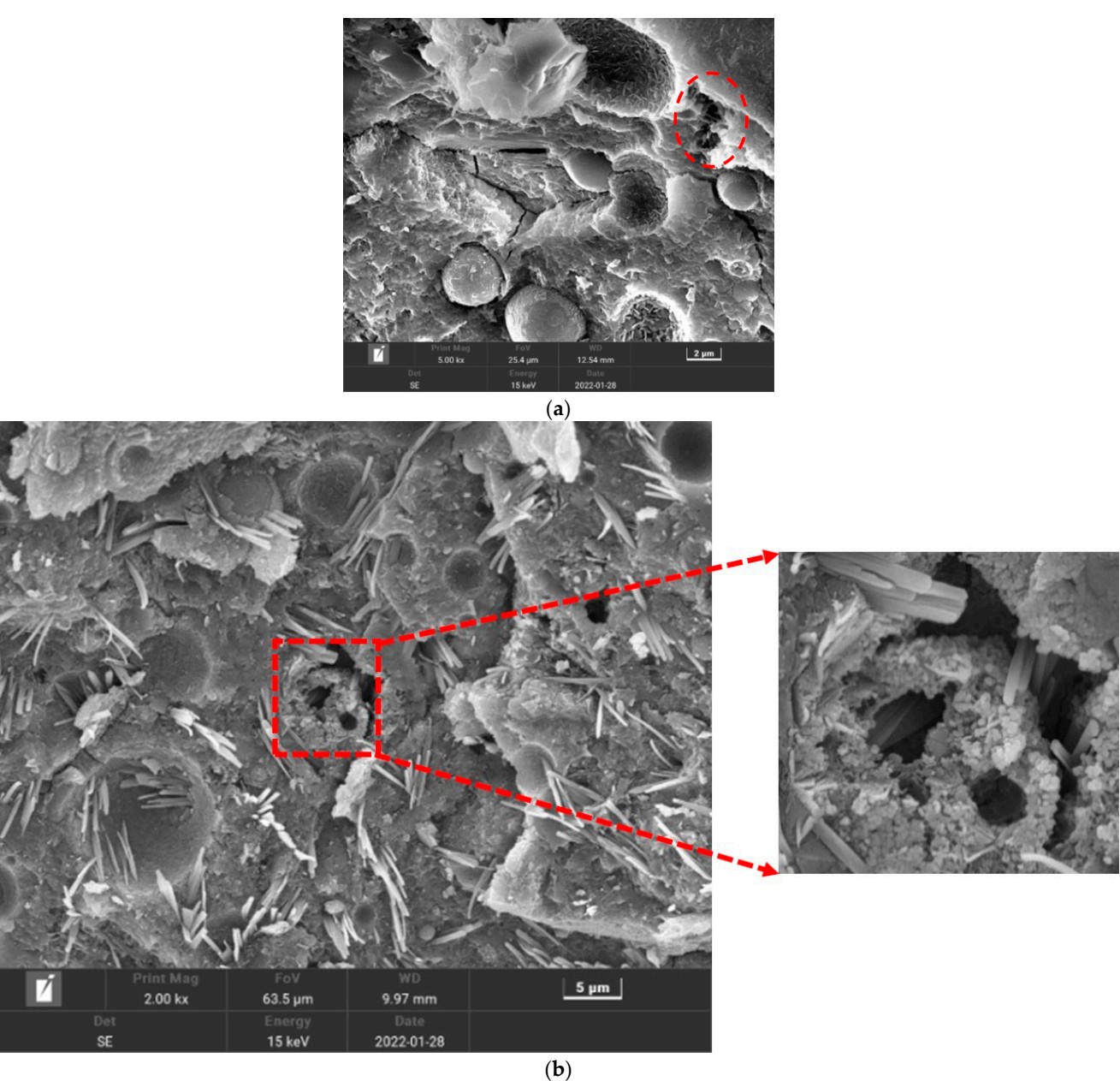

**Figure 11.** SEM results of RCASCC-50 at the end of 120 wet and dry cycles. (**a**) Erosion products and internal cracks. (**b**) Erosion products fill internal cracks.

### 3.6.3. Properties on the Micro Scale

The roughness of the interface of the self-compacting concrete with a higher replacement of RCA was generally higher, as shown in Figure 14. Substances from the outside world, such as sulfate ions, typically penetrate into concrete via the weak links of concrete. It has long been understood that the main weak point in concrete is the interfacial transition zone (ITZ) created between the aggregate and mortar [42]. The surface morphology of concrete ITZ is made up of a group of amorphous spherical particles with certain pore structures between them, according to He et al.'s research [39,43]. In the 3D illustration, the irregular spherical gullies somewhat matched the ITZ inside the concrete, where the aggregate and mortar were visible in contrast to the ITZ smoothness.

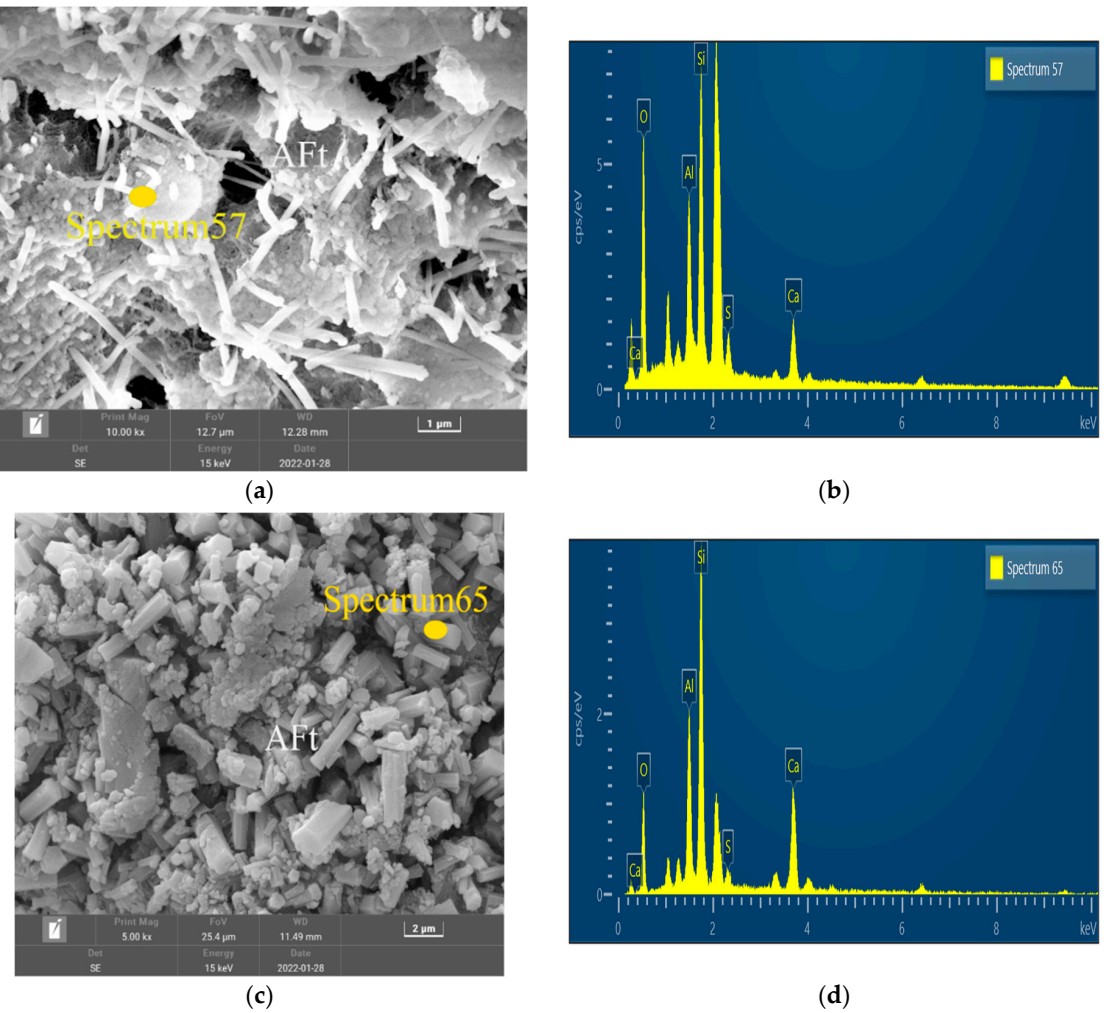

**Figure 12.** SEM results of RCASCC-25 after the end of 120 wet and dry cycles. (**a**) Pin cluster erosion products. (**b**) EDS energy spectrum point sweep analysis. (**c**) Pin-columnar erosion products. (**d**) EDS energy spectrum point sweep analysis.

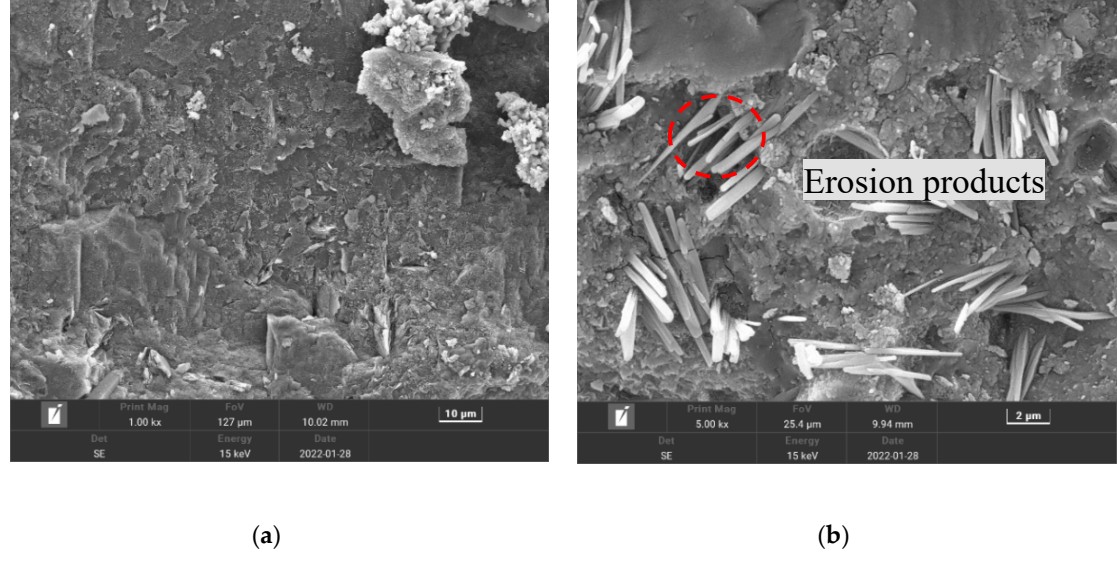

**Figure 13.** SEM results of RCASCC-100 at the end of 120 wet and dry cycles. (**a**) Surface structure of concrete after erosion. (**b**) Concrete surface erosion products.

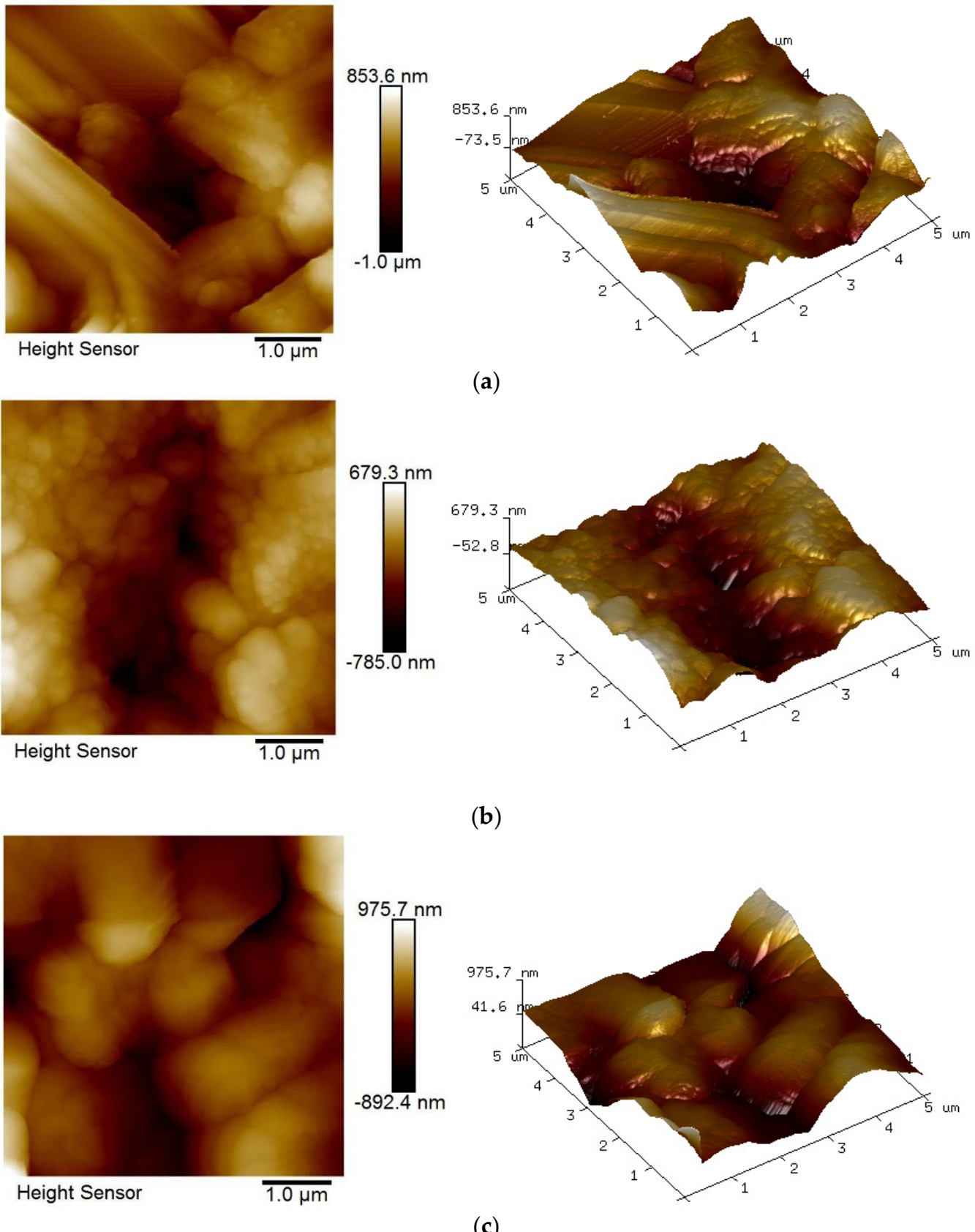

**Figure 14.** *Cont.*

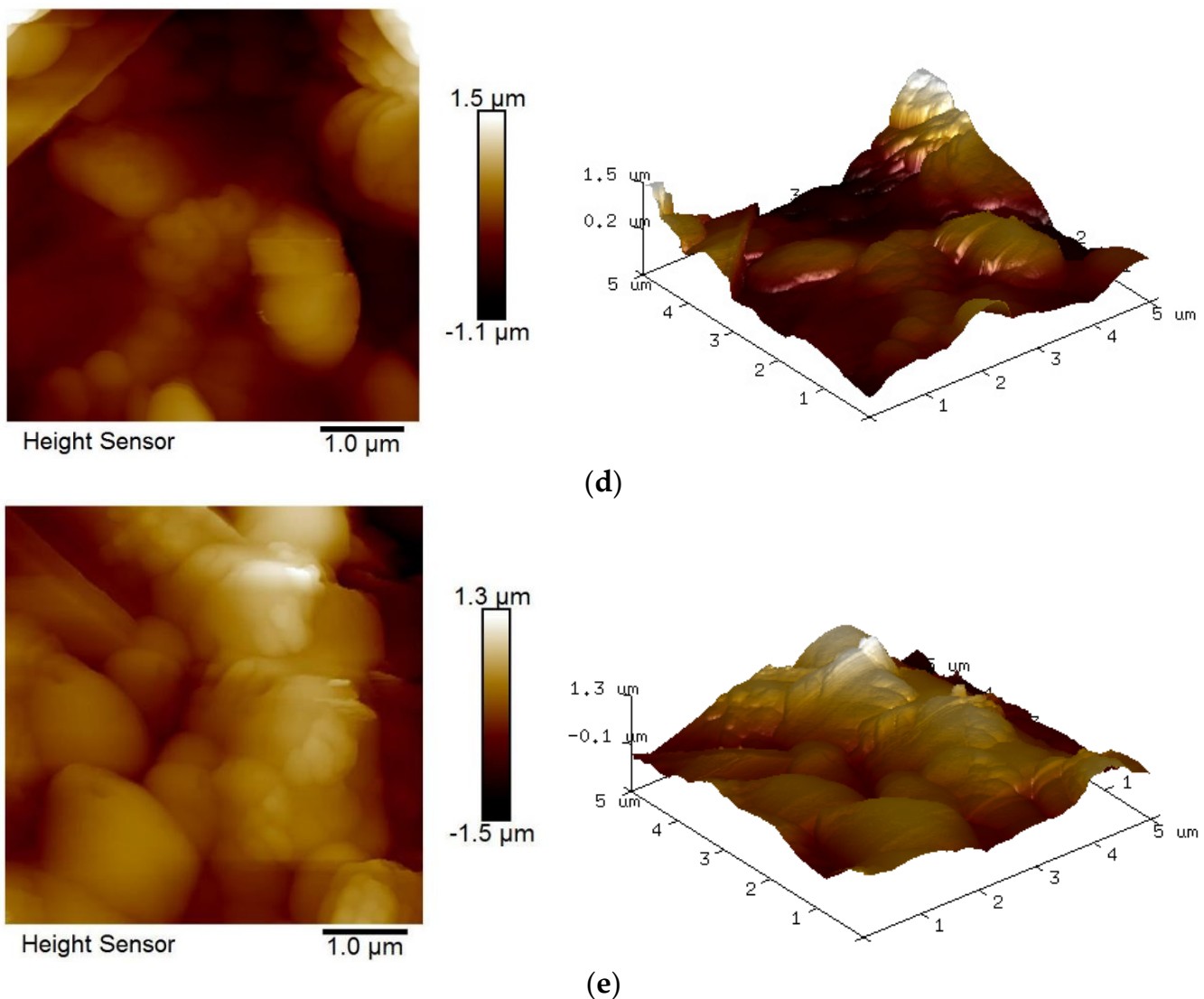

**Figure 14.** Changes in the transition zone of RCASCC interface under 120 wet and dry cycles. (**a**) RCASCC-0. (**b**) RCASCC-25. (**c**) RCASCC-50. (**d**) RCASCC-75. (**e**) RCASCC-100.

The graphical results in Figure 14a–e show how the internal ITZ of the self-compacting concrete varies in roughness according to the RCA replacement rate. The specimens with higher RCA replacement rates also clearly had higher roughness levels [41]. The overall was around 30% greater than the specimens with lower replacement rates, which laterally showed that the replacement rate of RCA would have an impact on the roughness of the specimens' internal ITZ. When coupled with cement and sand to harden in the concrete configuration, recycled coarse aggregate forms an ITZ surface that is rougher because it adheres to more old mortar and has a higher shape index during crushing. The primary cause was that, in areas where the ITZ structure was sparse, the concentration of $SO_4^{2-}$ significantly rose increased substantially with the number of the sulfate dry and wet cycles increased [23]. The swelling erosion products that were produced, including calcium alumina, filled the tiny gaps between the gel components and improved the microstructure of ITZ [43]. The outcome was a denser concrete construction. With the number of wet and dry cycles of sulfate erosion, the crystalline material in the ITZ structure of RCASCC steadily increased, and the pores between the hydration products gradually expanded [43]. The holes between the gel particles widened when more water molecules from the solution

entered the ITZ [44,45]. Later, as the initial cracks in ITZ grew and continued to form inside the concrete, the concrete's internal structure was destroyed.

## 4. Conclusions

In this study, the resistance of recycled aggregate self-compacting concrete to sulfate attack was investigated using the mass loss rate, relative dynamic modulus of elasticity, and corrosion resistance factor. Microscopic tests were used to observe and fen the recycled aggregate self-compacting concrete after erosion. The following are the primary conclusions:

(1)  With an increase in RCA, the compressive strength and 7/28 days strength ratio first rise and then fall, reaching their maximums at 50% of RCA. This finding suggests that concrete's compressive strength can be increased by adding the right amount of RCA.

(2)  The sulfate resistance of self-compacting concrete under the influence of wet and dry cycles is remarkable in response to the addition of RCA. Due to the high porosity of recycled aggregates, which creates additional channels for $SO_4^{2-}$ erosion and causes the gradual enlargement of primary and secondary fractures, the integration of increasing amounts of RCA significantly speeds up the pace of degradation at later stages.

(3)  $SiO_2$, AFt, and $CaCO_3$ are the byproducts of sulfate erosion that are discovered by XRD, and reliable results are acquired by microstructure SEM and EDS investigations. The old mortar stuck to the surface of RCA, as revealed by the AFM analysis of ITZ and the SEM analysis of RCA, significantly affects the roughness of ITZ inside RCASCC. The erosion products steadily rise with an increase in the RCA replacement rate, the ITZ becomes rougher, and the resistance to sulfate declines.

(4)  With an increase in the RCA replacement rate, the resistance of RCASCC to dry and wet cycles of sulfate falls in addition to the internal damage mechanism and the fundamental mechanical properties. Concrete's sulfate resistance rapidly declines, especially when the RCA replacement rate exceeds 50%. The RCA replacement rate should therefore not be higher than 50%.

## 5. Future Scope of Study

1.  The number of erosion cycles can be extended to 150 or even more than 300 to analyze the erosion resistance through lengthy cycles and conduct a more thorough investigation of durability.

2.  Other important features including loading, freeze–thaw, and chloride must be taken into account for further study in addition to the sulphate attack.

3.  CASCC may be taken into consideration for the addition of extra materials.

**Author Contributions:** S.L.: Formal analysis, Writing—Original draft, Investigation, and Writing—Review and editing; F.H.: Funding acquisition, Project administration, Conceptualization, Methodology, and Writing—Review and editing; S.Z.: Formal analysis and Writing—Review and editing; S.G.: Investigation and Data curation. G.Z.: Formal analysis and Writing—Review and editing. All authors have read and agreed to the published version of the manuscript.

**Funding:** This paper was supported by the National Natural Science Foundation of China (52168037).

**Institutional Review Board Statement:** Not applicable.

**Informed Consent Statement:** Not applicable.

**Data Availability Statement:** The research data to support the findings of this paper are available from the corresponding author upon request.

**Conflicts of Interest:** The authors declare no conflict of interest.

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
