# Peer review of "Study on Mechanical Properties and Erosion Resistance of Self-Compacting Concrete with Different Replacement Rates of Recycled Coarse Aggregates under Dry and Wet Cycles"

_applsci, doi:10.3390/app131911101_

Round 1
Reviewer 1 Report
Nice paper with an interesting study.
A few minor comments to be addressed before publication:
- In the introduction, double check the order of the statements. The second paragraph introduces SCC, while paragraph one already speaks about SCC with recycled aggregates.
- In general, check the short acronyms. There are a few typos (RAC instead of RCA, RSASS instead of RCASCC) and a few other typos.
- For both compression and erosion tests (with and without sulfate), how many samples are produced for each mix? If only one, what are the consequences on the reliability of the results (any quality control, reproducibility)? If more than one, please give average and standard deviation.
- Why not also considering Micro-Deval or Los Angeles test? What is the type of application targeted by this concrete?
Author Response
The response in word file.

Reviewer 2 Report
Overall, the English of the manuscript is very poor. The authors must proofread the article from a native English speaker before resubmission. Overall the manuscript is poorly written, the novelty of the study is missing, the detailed mechanism in explaining experimental results is missing, number and types of specimens used are not properly explained. Furthermore, the authors didn’t explain why and how they considered wet and dry cycles. More observation on the study is as follows;
· In abstract instead of emphasizing the results include the significance and experimental procedure.
· “To save construction resources, recycled coarse aggregates (RCA) partially replace natural coarse aggregates to make recycled coarse aggregate self-compacting concrete (RCASCC).” Correct the sentence, something is missing.
· Write as, “This paper studies the effect of coupled sulfate erosion on the mechanical property and durability of RCASCC with RCA replacement rates of 0%, 25%, 50%, 75%, and 100% under dry and wet cycles.”
· Line 13: delete “then”
· Line 23: “dry and wet cycles; sulfate resistant”
· Line 39. No need to mention Japanese scholar in the text.
· Author may consider citing
o https://doi.org/10.3390/su141811559
· Line 73: “Co.”
· Line 74: Remove extra space and at other locations in the manuscript.
· Line 76: local and domestic is the same thing.
· Line 94: Give space between text and citation. Similarly, at other locations.
· Line 107: It is suggested to show the WHY-3000 machine.
· In explaining Figure 1, only graph trend has been discussed. The authors are required to properly explain the mechanism of the behavior.
· Why graph in figure 1 show increasing, decreasing and increasing behavior.
· What is the minimum and maximum allowable compressive as per the used code/standard, mention in the graph in form of line?
· Add error bars in graphs.
· Table 4: avoid writing fcu/Mpa. Write as “fcu (MPa).
· Line 124: it is not mentioned in the text before that how many prismoids were casted.
· Line 124: How many cycles and on what basis these were selected?
· Heading given is “Mass loss rate” however graphs shows “Quality loss rate”
· Proper justification of increasing quality loss rate after around 30 cycles is not given in Figure 3 and 4.
· Why graph in figure 4, 5, 6 show increasing then decreasing behavior? A proper justification is missing in the manuscript.
· What is the purpose of calculating “relative dynamic modulus”?
· Heading 3.4: no need of adding long theoretical background from the literature.
· Table 5: Font size and type to be kept constant.
· Line 227: In text, it is written as “corrosion resistance coefficient” whereas, in the caption and figure it is written as “corrosion resistance factor”. Please clarify.
· Instead of showing researchers results in separate figure no 6. It is recommended to put the value in your own graph and present comparison.
· Remove un unnecessary gap between lines 303 and 305.
· How Figure 14 is drawn, readers might be interested in knowing.
· The author may use discussion of 3.5.1, 3.5.2 and 3.5.3 in explaining the mechanical behavior of RCASSA.
· Conclusion explained are very generic, the author must emphasize on concluding the novelty of the study.
· Quality of Figures 3 to 6 is very poor.
· Figure 7 is too small.
· References are not as per journal format.
Overall, the English of the manuscript is very poor. The authors must proofread the article from a native English speaker before resubmission.
Author Response
Thank you for pointing this out. This paper has been edited again. And Figure 2 ,4 to 7 have been changed as shown in manuscript lines 169, 209, 237, 269 and 286. The format of the references has been corrected.
How many cycles and on what basis these were selected?
120 cycles and base on the ‘standard for test methods of long-term performance and durability of ordinary concrete’ (GB/T 50082-2009).
What is the purpose of calculating “relative dynamic modulus”?
Thank you for pointing this. The purpose of calculating the relativistic modulus of elasticity is to characterize the ARSCC compactness on a macroscopic scale.
How Figure 14 is drawn, readers might be interested in knowing.
Figure 14 is a simple normalized treatment of AFM from a laboratory.
Reviewer 3 Report
The novelty of the manuscript needs to be highlighted
The recent literatures shall be added for discussion
The processing of RCA and its procedure is not clear
The details related to sample and procedure is missing for SEM/XRD
Check Table 3, Kg and some places CO2????
Write the limitations and scope for future work??
Add testing details and relevant pictures
Improve the results section with more appropriate results
Refer the following literature
Liu, J., Li, A., Yang, Y., Wang, X., & Yang, F. (2022). Dry–Wet Cyclic Sulfate Attack Mechanism of High-Volume Fly Ash Self-Compacting Concrete. Sustainability, 14(20), 13052.
Mathews, M. E., Anand, N., Kodur, V. K., & Arulraj, P. (2021). The bond strength of self-compacting concrete exposed to elevated temperature. Proceedings of the Institution of Civil Engineers-Structures and Buildings, 174(9), 804-821.
Author Response
The response in word file

Round 2
Reviewer 2 Report
Thank you for providing the revised manuscript. I appreciate the authors' efforts in thoroughly revising the entire document in response to the reviewer's comments.
Nevertheless, I have noted that a point-by-point response to my comments has not been included. This omission poses a challenge for the reviewer in identifying the specific changes made by the authors in accordance with the reviewer's suggestions. Therefore, I kindly request the authors to submit a detailed document that explicitly outlines all the corrections made in response to the reviewer's comments.
This comprehensive document will serve to enhance transparency and facilitate a clear understanding of the modifications made within the context of the reviewer's feedback. I believe that providing this detailed sheet will help streamline the review process and ensure that the manuscript addresses all relevant points raised during the initial evaluation.
I value the authors' dedication to improving their work and look forward to receiving the requested document. Thank you for your cooperation.
N/A
Author Response
The report upload in the word file.

Reviewer 3 Report
The manuscript has been improved
Author Response
Thank you.
Round 3
Reviewer 2 Report
The quality of the paper has been improved. On page 12, it is Figure 9 not 1. Please correct.
English of the manuscript seems to be improved.